# Camera-Based Blind Spot Detection with a General Purpose Lightweight Neural Network

**Yiming Zhao, Lin Bai, Yecheng Lyu** and **Xinming Huang** *

Department of Electrical and Computer Engineer, Worcester Polytechnic Institute, Worcester, MA 01609, USA; yzhao7@wpi.edu (Y.Z.); lbai2@wpi.edu (L.B.); ylyu@wpi.edu (Y.L.)
* Correspondence: xhuang@wpi.edu

**Abstract:** Blind spot detection is an important feature of Advanced Driver Assistance Systems (ADAS). In this paper, we provide a camera-based deep learning method that accurately detects other vehicles in the blind spot, replacing the traditional higher cost solution using radars. The recent breakthrough of deep learning algorithms shows extraordinary performance when applied to many computer vision tasks. Many new convolutional neural network (CNN) structures have been proposed and most of the networks are very deep in order to achieve the state-of-art performance when evaluated with benchmarks. However, blind spot detection, as a real-time embedded system application, requires high speed processing and low computational complexity. Hereby, we propose a novel method that transfers blind spot detection to an image classification task. Subsequently, a series of experiments are conducted to design an efficient neural network by comparing some of the latest deep learning models. Furthermore, we create a dataset with more than 10,000 labeled images using the blind spot view camera mounted on a test vehicle. Finally, we train the proposed deep learning model and evaluate its performance on the dataset.

**Keywords:** squeeze-and-excitation; residual learning; depthwise separable convolution; blind spot detection

---

## 1. Introduction

In the 2012 ILSVRC competition, deep convolutional neural network designed by Hinton et al. achieved the lowest error rate of 15.3% that is 10.8% better than the runner up [1]. The large and complex classification dataset in ILSVRC with 1000 object categories is widely regarded as a benchmark to evaluate different machine learning models [2]. This milestone achievement attracted many researchers to the field of deep neural networks. In the following years, adaption of new neural network structures continually pushed the error rate lower. In ILSVRC-2014, GoogLeNet achieved 6.67% error rate using inception module [3]. A series of inception modules can find the optimal local construction by concatenating output from convolution kernels in different sizes [4,5]. In the same year, VGG model was published with outstanding performance results, which quickly became one of the most popular structures [6]. In ILSVRC-2015, the invention of residual error in ResNet made it possible to train a very deep network with more than 100 convolution layers [7]. In ILSVRC-2017, researchers from University of Oxford even achieved 2.25% error rate with squeeze-and-excitation module [8].

The success of deep convolutional neural network in image classification stimulates the research interest of solving many other challenging computer vision problems. For object detection tasks, some models like YOLO [9,10] and SSD [11] directly transformed this task to a regression problem with neural network. The R-CNN series model [12–14] replaced the traditional histogram of gradients (HOG) with Selective Search and SVM method [15] using deep neural network. Mask R-CNN [16] can directly generates pixel level semantic segmentation with bounding box. More complex tasks usually

require the extraction of more elaborated and accurate information from images. Other researchers proposed various convolutional kernels, such as deformable convolution [17] which can change the shape of kernels or dilated convolution [18] which can capture a larger range of information with the same kernel size.

For an embedded system with limited computing capacity, the heavy computational cost of very deep neural network is prohibitive. Methods to increase the training and inference speed by reducing the parameters and operations become an important topic lately. Many recent models such as Xception [19] and MobileNet [20] designed the neural networks based on depthwise separable convolutions. This module can dramatically reduce the number of parameters without losing too much accuracy. Furthermore, ShuffleNet [21] utilized less parameters by shuffling and exchanging the output after group convolutions.

However, neural networks in all the aforementioned models still contain lots of layers. The MobileNetV2 [22] released recently has 17 blocks which include 54 convolution layers in total. The question is—do we really need such a large network to solve a specific real-world problem? If the task is not that complex and we ought to avoid the very deep structure, how should we adapt the deep learning models? Do we still need residual learning? In order to answer those questions, we propose a network structure based on AlexNet [1]. After the first convolution layer, there are four identical blocks. For each block, we consider different combinations of the latest deep learning models including residual learning, separable depthwise convolution and squeeze-and-excitation. The final model is chosen by comparing the evaluation accuracy and computing cost.

Finally, we implement our proposed model for camera-based blind spot detection. Blind spot detection is very important to driving safety. However, the radar based system is relatively expensive and has a limited ability for complex situations. There are few works focusing on camera based blind spot detection and the existing publications are largely based on artificial features or traditional signal processing methods [23–26].

There are two main contributions in this paper. One is that we combine depthwise separable convolution, residual learning and the squeeze-and-excitation module together to design a new block. Compared with VGG block, deep neural network composed of the proposed new block can achieve similar performance but with significantly less parameters when evaluated on CIFAR-10 dataset and our own blind spot dataset. More importantly, we present a complete solution to camera-based blind spot detection from experimental setup, data collection and labelling, deep learning model selection, and performance evaluation.

## 2. Related Work

Our goal is to design a deep neural network model that can solve real world problems without involving too many layers. The basic principle to design a lightweight neural network is reducing the model size without losing too much accuracy. So we briefly introduce the latest works on network size reduction. Then, we revisit three deep learning modules considered in this paper.

### 2.1. Existing Work on Reducing Network Size

Most networks are built with 32-bit floating point weights, so an intuitive reduction technique is to quantize the weights in fixed-point representation. By using 16-bit fixed-point representation in stochastic rounding based CNN training, the model memory usage and operations were significantly reduced with little lost on classification accuracy [27]. The extreme case for weight representation is the binary notation. Some researchers directly applied binary weights or activations during the model training process [28,29].

Model compression is another effective approach. Almost 30 years ago, the universal approximation theorem of neural network stated that simple neural networks can represent a wide variety of nonlinear functions when given appropriate parameters [30]. It was reported using a shallow network to mimic the complex, well-engineered, deeper convolutional models [31]. For a complex

network, discarding redundant nodes is another way to reduce the model size. A recent work combined model pruning, weights quantization and Huffman coding to achieve better performance [32].

### 2.2. Deep Learning Module Revisit

This paper is aiming at designing network structure to reduce the computational cost. Separable depthwise convolution can dramatically decrease the number of parameters and operations of the network. Since separable depthwise convolution may result in the loss of accuracy, we consider adding residual learning and squeeze-and-excitation module. These two modules require little additional computing cost, but they are capable of improving accuracy.

#### 2.2.1. Separable Depthwise Convolution

Depthwise separable convolution became popular recently for mobile devices [19–22,33]. Although there is difference among several existing works, the core idea is the same. Compared with standard $3 \times 3$ convolution in VGG, depthwise separable convolution does channel-wise convolutional calculations with $3 \times 3$ kernels. Then standard $1 \times 1$ convolutions are applied to integrate information for all channels. Let us assume the number of input channel is $M$ and the number of output channel is $N$. Standard $3 \times 3$ convolutions require $M \times 3 \times 3 \times N$ parameters. The depthwise separable convolutions only need $M \times 3 \times 3 \times 1 + M \times 1 \times 1 \times N$ parameters which are much less than the standard $3 \times 3$ convolutions.

#### 2.2.2. Residual Learning

When researchers made the network deeper and deeper, they encountered an unexpected problem. As the network depth increases, accuracy gets saturated and then degrades rapidly. Residual learning solves this problem with an elegant yet simple solution [7]. In deep residual learning framework, as shown in Figure 1, the original mapping $F(x)$ is recasted into $F(x) + x$, which is the summation of original mapping and the identity mapping of input. This simple solution magically makes it possible to train a very deep neural network with only a small increase of computation. Hence, residual learning quickly became a popular component in the latest deep learning models. DenseNet is also a helpful way to train very deep neural networks [34]. However, the concatenating operation will greatly increase computational cost and parameters. Thus we do not consider DenseNet in this paper.

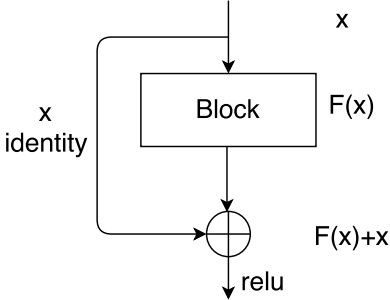

**Figure 1.** Residual learning on an existing block.

#### 2.2.3. Squeeze-and-Excitation

Based on the squeeze-and-excitation (SE) block, SE network won the first place of the classification task in ILSVRC 2017 with the top-5 error 2.25% [8]. SE module can re-weight each feature map by imposing only a small increase in model complexity and computational burden.

In Figure 2, we show how SE module operates with an existing block. Let us assume the output tensor is $A$ and the shape of A is $W \times H \times C$. Then, the global average operation maps each feature map into one value by calculating the average on each feature map.

$$\hat{A}_k = \frac{1}{W \times H} \sum_{i=1}^{W} \sum_{j=1}^{H} A_{ijk}$$

After that, the first fully connected layer performs the squeezing step by $\frac{C}{r}$ units, and the second fully connected layer does the excitation step by $C$ units. r is the reduction ratio in SE module, which can decide the squeeze level of the module and affect the number of parameters. Finally, a sigmoid activation layer transforms it to probability and does the multiplication with original tensor $A$.

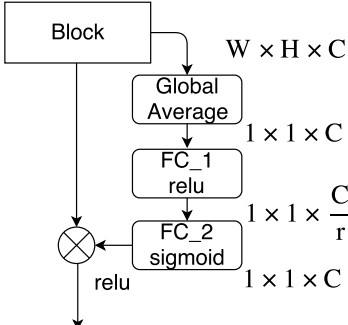

**Figure 2.** Squeeze-and-excitation on an existing block.

## 3. Investigation of Deep Learning Models

As we mentioned earlier, researchers designed many new network modules to empower the deep learning in recent years. However, an embedded computing platform can only handle the computation load of a model with a few layers. In this case, how should we choose those modules? Here, we hold the same network structure but change the setting of the building block. We propose four different blocks, corresponding to four different neural networks. In Figure 3a, we show the entire network structure. In order to keep it simple and intuitive, we use a VGG-like structure. We begin with one standard convolution layer to extract information from the input data. The kernel size is decided by the shape of the input tensor. For CIFAR-10 dataset, the input shape is $(32, 32, 3)$, so we use $3 \times 3$ kernel with stride 1. For blind spot detection dataset, the input shape is $(128, 64, 3)$, so we use $5 \times 5$ kernel with stride 2. Then, a max pooling is used to compress the information. The main body of the structure consists of four identical blocks. If we choose a VGG block, then there should be four VGG blocks in the main part of the network. The output part contains one average pooling and two fully connected layers. Finally, an activation layer shows output probability of each category. We use sigmoid for two-class problems and softmax for multi-class problems. In Figure 3, we show the details of four different blocks. In Figure 3b, we can see the VGG block has a standard convolution followed by batch normalization and relu function. We set this block as the baseline. Next, we use depthwise separable convolution to replace the standard convolution. This modification reduces the number of parameters significantly. Furthermore, we separately equip the new convolution with residual learning as in Figure 3c to form the Sep-Res block, or with squeeze-and-excitation as in Figure 3d to form Sep-SE block. Finally, we combine those three parts together to ensemble the Sep-Res-SE block, as in Figure 3e. By comparing the results of Sep-Res block and Sep-Res-SE block, we can evaluate the performance of squeeze-and-excitation. By comparing the results of Sep-SE block and Sep-Res-SE block, we can evaluate the performance of residual learning.

Residual learning requires the output tensor in the same size of the input tensor, so we need to keep a constant number of channels in the block. In this paper, we solve this problem by adding one $1 \times 1$ standard convolution at the beginning of the block as in Figure 3c–e. This bottleneck convolution

can increase the number of channels as needed. When we train squeeze-and-excitation module, we find it is harder to converge. So we add batch normalization before the multiplication to help it converge faster.

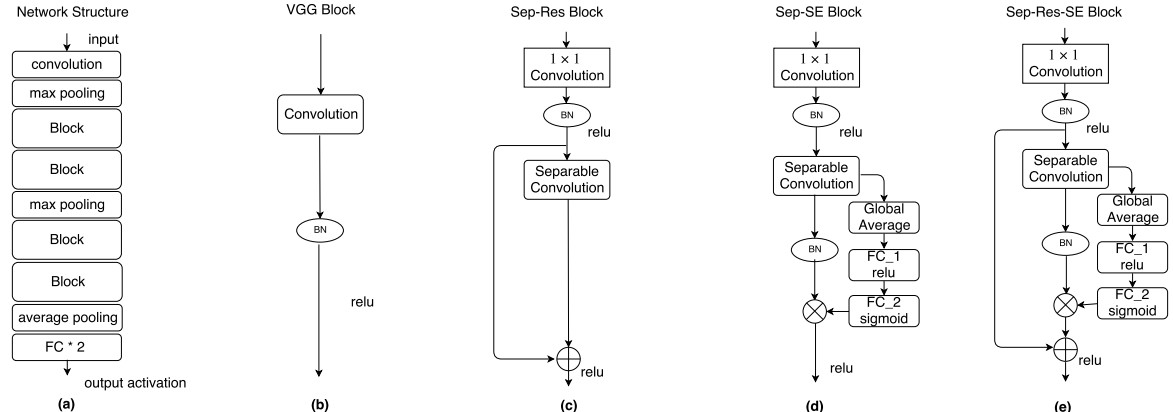

**Figure 3.** We show the structure of our neural network in (**a**). The first convolution layer extracts information from input data and the two fully connected layers at the bottom gradually change the output size to class number. There are four blocks in the main body of the network; we can put different settings in all those four blocks to see how to design network can perform better. (**b**) is a standard VGG Block, a convolution followed by batch normalization and relu. (**c**) is a Sep-Res Block, we replace standard convolution with separable depthwise convolution and add residual learning module on it. The first $1 \times 1$ convolution increase channels to guarantee the output shape is the same as input shape for residual. (**d**) is a Sep-SE Block, we replace the residual learning module by squeeze-and-excitation module. In (**e**), we combine all those parts together to form Sep-Res-SE Block.

## 4. Experiments and Evaluation

### 4.1. Datasets

#### 4.1.1. CIFAR-10

The CIFAR-10 dataset is a popular dataset for evaluation of machine learning methods. It contains 60,000 $32 \times 32$ color images in 10 different categories. The low resolution, a few categories and sufficient samples in each category make this dataset suitable for evaluating the performance of our proposed models.

#### 4.1.2. Blind Spot Detection

In this subsection, we discuss the procedures to transform the blind spot detection to a machine learning problem. We draw the blind spot region in Figure 4. The region consists with four $4 \text{ m} \times 2 \text{ m}$ rectangles. Driver should be alerted if any car enters this region. So we model blind spot detection as a Car or No-Car two-class classification problem. No-Car class indicates there is no vehicle in the blind spot region, so it is safe for lane changing. Car class means there is at least part of a vehicle in the blind spot region, thus driver should not change lanes. Our test vehicle is a Lincoln MKZ equipped with high-resolution Sekonix cameras. We mount the blind spot camera on the side of rooftop with 45 degrees facing backward, and the position of blind spot camera is shown in Figure 5a. In order to capture information better, we create a 3D region with 2 m in height recorded by camera in Figure 5b. Before feeding the training image into models, we preprocess original image by clipping the blind spot region and resize it to $128 \times 64$. Figure 5c is the input image of Figure 5b after preprocessing.

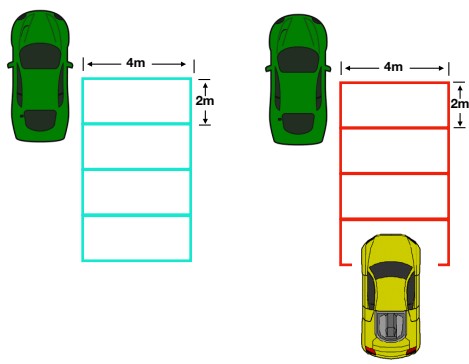

**Figure 4.** Bird's-eye view of blind spot region.

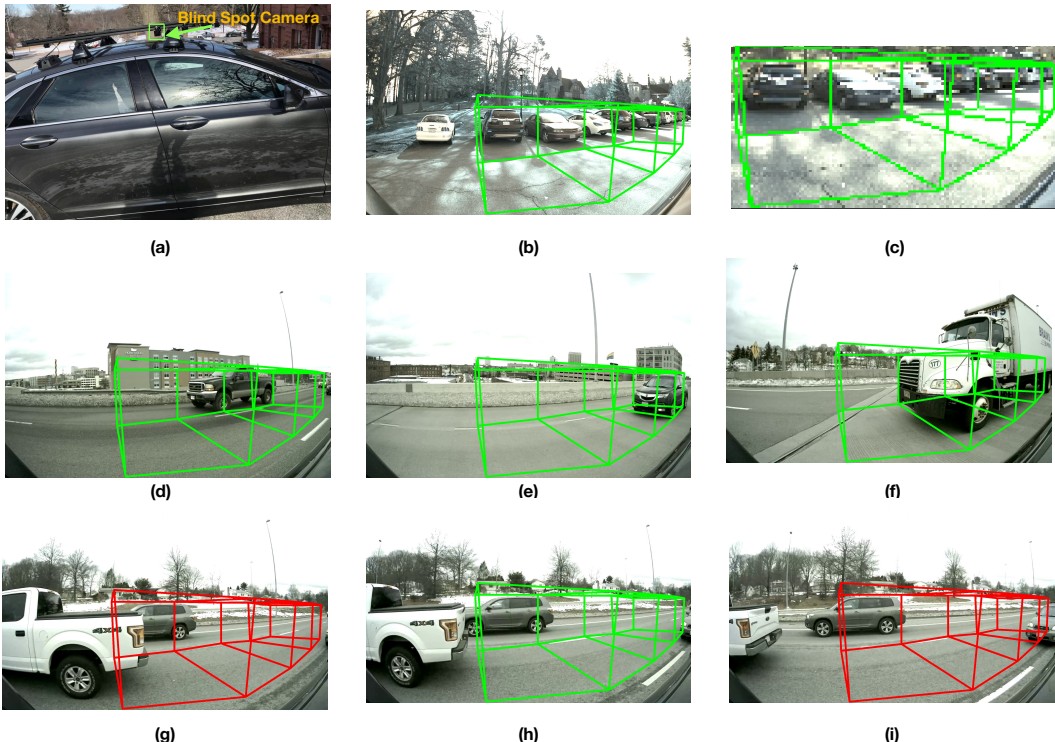

**Figure 5.** We show the position of our camera in (**a**). (**b**) is the blind spot region in the view of camera. (**c**) is the training image after preprocessing (**b**). (**d**) is an example of No-Car class in training data, (**e**,**f**) are examples of Car class in training data. (**g**–**i**) are examples of the final output of our blind spot detection system. If the model finds a car, it will alarm the driver by changing the color of the box. Our model is fairly accurate and can exactly account for the blind spot region.

We record several videos when driving on highways and combine them together into one large video. Then, we split the video as training video and test video. For training video, we choose one out of every five frames as the training image. For test video, we use all the frames without sampling as the test dataset. Next, we draw the blind spot region on the training dataset and label them manually. As an example, Figure 5d belongs to No-Car class since the vehicle did not stay in the prescribed 3D region. Figure 5e,f both belong to the Car class since at least part of a vehicle appears in the blind spot region. In total, we obtained 8336 images of the No-Car class and 2184 images belonging to the Car class. The imbalanced dataset would limit the performance of machine learning models and some methods were developed to solve this problem [35,36]. Here we take a simple task-specific policy to balance the dataset. For No-Car class, we discard 3336 similar images which just contain road surface. For Car class, we double the images each with a vehicle occupying two or more rectangles in the blind spot region. Finally, we obtained 5000 images in No-Car class and 3874 images in Car class.

### 4.2. Experiment Setting

In this part, we discuss the details of model setting. All the models in this paper share the same setting for the first layer, that is a standard convolutional layer with 64 channels. For CIFAR-10 dataset, it has $3 \times 3$ convolutional kernel with stride 1. For Blind Spot Detection dataset, it has $5 \times 5$ convolutional kernel with stride 2. The successively repeated four blocks in each model have the same setting for both datasets. All the standard convolutional layers and separable convolutional layers in each of those four blocks use $3 \times 3$ kernel with 1 stride. There are 128 channels in the first two blocks, and 256 channels in the last two blocks. The squeeze factor r is 16. All the pooling layers have $2 \times 2$ kernel with stride 2.

For each dataset, we use the same learning rate and batch size to make a fair comparison among four different neural networks. For CIFAR-10, we set *learning_rate* = 0.001, *batch_size* = 64, choose Adam [37] as the optimizer and train all four models for 100 epoch. For blind spot dataset, we set *learning_rate* = 0.0001, *batch_size* = 64, choose Adam as the optimizer and train all the four models for 30 epoch.

### 4.3. Results

In Table 1, we compare the test accuracy of all four models on two datasets. The Sep-Res-SE model that combines depthwise separable convolution, residual learning and squeeze-and-excitation performs better than the other two models. The VGG Block still has slightly higher accuracy than Sep-Res-SE Block. However, test accuracy is not the only factor that we should consider for the real-time problem. Inference speed and memory cost are also important for an embedded system. We list the inference speed for each model on blind spot dataset with Nvidia Quadro p6000 in Table 1. The model with VGG block requires nearly twice as much time to handle one image when comparing to the other modules. In fact, inference speed is decided by the amount of operations and the memory cost is decided by the number of parameters. In Table 2, we show the number of parameters and operations in the first block of each model. Since four repeated blocks comprise the main body of the model, the parameters and operations in one block can clearly show the computational cost of each model. From Table 2, the VGG model that achieves higher test accuracy requires twice or more parameters and operations. Compared with standard convolution, models equipped with depthwise separable convolution require far fewer parameters and operations. By combining residual learning and squeeze-and-excitation, we can compensate the test accuracy of depthwise separable convolution with slightly more parameters and operations but achieve similar accuracy comparable to the VGG Block.

Therefore, the combining of depthwise separable convolution, residual learning and squeeze-and-excitation is the best tradeoff between accuracy and cost. Subsequently, we apply the trained neural network model to our test vehicle with cameras installed. Figure 5g–i are examples from the real application. It shows that the proposed method can detect the car in the blind spot region effectively with no confusion by the vehicles outside of the region. Our test video posted online shows that the proposed model nearly detected all the cars in the blind spot region. A few mistakes occurred owing to the shadow of the bridge casted on the surface of the road. The model can be further improved by adding more training data.

MobileNet is one of the well-known deep learning structures for mobile devices. For comparison, we also implemented MobileNetV2 which is the recent version of MobileNet. We set the *learning_rate* = 0.0001, *batch_size* = 64, choose Adam as the optimizer and train it for 100 epoch. We show the comparison results in Table 3. Since the number of model operations are strongly affected by the input tensor size and number of filters, the idea of MobileNetV2 is to downsample the image size quickly and to use less filters in the top layers. However, MobileNetV2 requires many more layers to keep up the performance. As in Table 3, MobileNetV2 has less operations but more parameters comparing with Sep-Res-SE. We test both MobileNetV2 and Sep-Res-SE in the same environment. The MobileNetV2 get 95.35% test accuracy which is lower than the Sep-Res-SE. The inference speed of MobileNetV2 is

also slower than Sep-Res-SE. Although we believe MobileNetV2 may be able to achieve similar or even better results after fine tuning the training process, it is not efficient to use a very deep neural network with large memory cost for the blind spot detection problem.

**Table 1.** This table show the test accuracy on two datasets with four different neural networks and the inference speed on blind spot dataset.

| | Test Result | | |
|---|---|---|---|
| **Network Block Type** | **CIFAR10** | **Blind Spot** | **Inference Speed per Image** |
| VGG Block | 0.8829 | 0.9801 | 0.00259s |
| Sep-Res Block | 0.8554 | 0.9737 | 0.00159s |
| Sep-SE Block | 0.8575 | 0.9701 | 0.00166s |
| Sep-Res-SE Block | 0.8730 | 0.9758 | 0.00169s |

**Table 2.** This table shows the number of parameters and operations of the first block in each model. We count both multiplication and add(Multi-Add) as the operation.

| | CIFAR10 | | Blind Spot Detection | |
|---|---|---|---|---|
| | **Params** | **Multi-Add** | **Params** | **Multi-Add** |
| VGG Block | 73.7k | 37.8M | 73.7k | 302.5M |
| Sep-Res Block | 25.7k | 13.3M | 25.7k | 106.2M |
| Sep-SE Block | 33.9k | 13.4M | 33.9k | 106.9M |
| Sep-Res-SE Block | 33.9k | 17.6M | 33.9k | 140.5M |

**Table 3.** This table shows the comparison with the model proposed in this paper with MobileNet which is a famous neural network structure for mobile device.

| Model Comparison on Blind Spot Detection Dataset | | | | |
|---|---|---|---|---|
| **Model** | **Params** | **Multi-Add** | **Accuracy** | **Inference Speed per Image** |
| Sep-Res-SE | 143.4k | 420M | 0.9758 | 0.00169s |
| MobileNetV2 | 3.4M | 48.9M | 0.9535 | 0.00488s |

## 5. Conclusion and Discussion

In this paper, we discuss how to design a neural network with only a few layers for real-time embedded applications, such as blind spot detection. Usually higher accuracy requires deeper model and better computational cost. By using depthwise separable convolution, we dramatically reduce the model parameters and operations. Then, we add residual learning and squeeze-and-excitation module to compensate the loss of accuracy with only a small increase of parameters. Compared with VGG block, the Sep-Res-SE block, combining depthwise separable convolution, residual learning and squeeze-and-excitation can achieve similar detection accuracy with far fewer parameters and operations. We recommend this model as the best tradeoff between accuracy and cost.

We also present a complete solution to camera-based blind spot detection. We successfully solve this problem by building a machine learning model from labeling dataset. However, we have not yet considered all the situations with different roads and weather conditions due to the additional workload of gathering and labeling data. Moreover, if geo-information is provided by sensors like Inertial Measurement Unit (IMU), the model can be further improved for sloped road by adjusting the blind spot region.

**Author Contributions:** Conceptualization, Y.Z. and L.B.; methodology, Y.Z.; formal analysis, Y.Z.; data curation, Y.L. and Y.Z.; writing—original draft preparation, Y.Z.; writing—review and editing, X.H.; supervision, X.H.; project administration, X.H.; funding acquisition, X.H.

**Funding:** This research was funded by National Science Foundation grant number 1626236.

**Conflicts of Interest:** The authors declare no conflict of interest.

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
