# Peer review of "Camera-Based Blind Spot Detection with a General Purpose Lightweight Neural Network"

_electronics, doi:10.3390/electronics8020233_

Round 1

Reviewer 1 Report

This paper presents neural networks for object detection in images from cameras that cover the blind spots of a car. The NNs are designed relatively small in size in an aim to keep the computational and storage requirement.

1.    The details of NN architectures are not provided or not clear from the description, e.g., the number of layers, number and size of filters, stride in each layer etc.

2.    Does the proposed method consider adjusting the region of interest in an image when the car is taking a turn or driving on a sloped road? 

3.    Little details about the car and camera are given in the paper. Please provide the models of the vehicle and cameras used, and their location and orientation. 

4.    It seems from the figures that the cameras used for the experiment are fisheye cameras. Are the images trained on a raw image or corrected image taking into account intrinsic calibration parameters and matrices? If a raw image is used, please explain the reasons.

5.    It appears this method considers only vehicles for obstacle detection. What about other dynamic objects such as bikes and pedestrian?

6.    Please add the number of parameters and FLOP for MobileNetV2 in Table 2 and compare computational and storage requirements against Sep-Res-SE.

Author Response

Point 1: The details of NN architectures are not provided or not clear from the description, e.g., the number of layers, number and size of filters, stride in each layer etc.

Response 1: Thanks for your opinion and sorry for the unclear. The details of NN is crucial for people to repeat the result. In the Experiment section, we add a new subsection named Experiment Setting which list all those details. And we also move the training details, like epoch and learning rate into this subsection. We believe the new subsection can make things in this point clear and help people to repeat our model.

Point 2: Does the proposed method consider adjusting the region of interest in an image when the car is taking a turn or driving on a sloped road? 

Response 2: For the turning situation, since the blind spot region is decided by car, we think the region should be fixed in terms of the car. However, for the sloped road, we agree adjusting the region is more useful. But adjusting region according to slope needs more sensors, like IMU to provide the geo information, so we still didn’t cover it in this paper. We think this point is very good, so we add the response into the last Conclusion and Discussion section in the paper.

Point 3: Little details about the car and camera are given in the paper. Please provide the models of the vehicle and cameras used, and their location and orientation. 

Response 3: Thanks for the reminding. We add the information of the car in the Dataset section before the introduce of the camera.

Point 4: It seems from the figures that the cameras used for the experiment are fisheye cameras. Are the images trained on a raw image or corrected image taking into account intrinsic calibration parameters and matrices? If a raw image is used, please explain the reasons.

Response 4: We train the model based on corrected image which seems a standard part for most image projects. But we didn’t mention it in the paper. Since we think deep learning is a data-based method, the correction shouldn’t affect the efficiency of the model. 

Point 5: It appears this method considers only vehicles for obstacle detection. What about other dynamic objects such as bikes and pedestrian?

Response 5: Thanks, this is a very good point for project intention. We have an autonomous driving project, so when we start designing this bind spot paper, we mainly focus on how to help the car changing lane safety. In some other blind spot detection paper for truck like [1], they mainly consider the pedestrian or cyclist, since they are trying to avoid damage to the surroundings. However, the method in our paper can be easily extended to cover more situations like bikes or pedestrian. Since we transfer the blind spot detection to a classification problem, people just need to capture more data and add more classes to meet their requirement. We also add the response here into the last Conclusion and Discussion section.

Point 6: Please add the number of parameters and FLOP for MobileNetV2 in Table 2 and compare computational and storage requirements against Sep-Res-SE.

Response 6: Thanks for the suggestion. Actually, the comparison in Table 2 is based on first block. In order to make a better comparison, we create a new Table 3. In Table 3, we compare the param/operations/accuracy/speed of MobileNet against Sep-Res-SE. And we also discuss further the comparison results in the paper. We believe this is a more clear way to make the comparison.

[1] Van Beeck, K.; Goedemé, T. The automatic blind spot camera: a vision-based active alarm system. European Conference on Computer Vision. Springer, 2016, pp. 122–135.

Reviewer 2 Report

In this paper, it proposed to combine depthwise separable convolution, residual learning and squeeze-and-excitation to build new deep learning models for real-time applications. It achieves similar detection accuracy with less parameters and operations.

To further improve the manuscript, the authors may address the following comments:

1. The intuition of the proposed method is lacking in some aspects.

a. In the proposed method, it combines depthwise separable convolution, residual learning and squeeze-and-excitation for a new deep learning model. Firstly, it is important to include the discussion why these components are useful to build a deep learning model for real-time applications. What are the essential properties of these components?

b. Why the combination of these components can significantly reduce the model size? Without theoretical analysis or strong intuitions, the proposed structure may look like random trials to achieve fast computation.

2. The section of ‘Related Work’ is not proper.

a. The related works are not adequately reviewed. To achieve fast computation and reduced model parameters, many model compression methods are proposed for deep learning. Another research directions is to alter the model structure based on the analysis of the model performance for further model merge or split to reach the goal of model size reduction. However, none of these papers are reviewed. In the current manuscript, only Xception[19], MobileNet[20] and ShuffleNet[21] are included in the discussion. This is not enough.

b. I suggest re-organize the paper. In Section 2, it should review papers in the aforementioned research directions. Add a new sub-section such as deep learning method revisit to review three methods: depthwise separable convolution, residual learning and squeeze-and-excitation, since they are the main components in your model.

3. The paper title is not appropriate. Since the paper aims to propose a new deep learning structure for real-time applications, the current title does not related to the paper goal at all. In fact, the proposed method is very general and can be applied to any real-time problems, the current title is misleading that the paper is just for blind spot detection problem.

4. The experimental results are not convincing enough.

a. In Fig.4, it depicts the blind spot region. However, In Fig.5, sub-figures do not comply with the definition of blind spot in Fig.4. Why blind spot regions can appear in front of cars? In addition, (e) and (f) are examples of CAR class. However, there is no car in the blind spot region. Why they are CAR class?

b. In Table 1 and Table2, we can see the model parameters and the operations of MADD is significantly reduced. However, the inference speed is just 0.001 second. The impact of the proposed method is not significant.

c. It does not compare other model compression methods. Only VGG block is compared in the experiments. This is not convincing enough.

d. In the proposed method, three model settings are used: Sep-Res Block, Sep-SE Block and Sep-Res-SE Block. Which one is better? A discussion section should be provide to compare the results for further conclusion.

5. There are many grammar errors in this manuscript. For example, in the abstract, ‘deep learning method that accurately detect’ should be ‘deep learning method that accurately detects’, ‘a series of experiments is conducted’ should be ‘a series of experiments are conducted’.

Author Response

Point 1: The intuition of the proposed method is lacking in some aspects.

a. In the proposed method, it combines depthwise separable convolution, residual learning and squeeze-and-excitation for a new deep learning model. Firstly, it is important to include the discussion why these components are useful to build a deep learning model for real-time applications. What are the essential properties of these components?

b. Why the combination of these components can significantly reduce the model size? Without theoretical analysis or strong intuitions, the proposed structure may look like random trials to achieve fast computation.

Response 1: Thank you very much for the opinion. We agree it’s crucial to make the designing intuition more clear. In general, depthwise separable convolution is a popular way to reduce the operations and parameters without losing too much accuracy. Residual learning and squeeze-and-excitation can not reduce the parameters and operations, but they can improve the model performance with a very little increase of the parameters and operations. So the Sep-Res-SE model we proposed can reduce the parameters by using depthwise separable convolution and can compensate the model performance by using residual learning and squeeze-and-excitation. We apologize for the unclear in the paper. We add the response here in the new Deep Learning Module Revisit  subsection within Related Work part.

Point 2: The section of ‘Related Work’ is not proper.

a. The related works are not adequately reviewed. To achieve fast computation and reduced model parameters, many model compression methods are proposed for deep learning. Another research directions is to alter the model structure based on the analysis of the model performance for further model merge or split to reach the goal of model size reduction. However, none of these papers are reviewed. In the current manuscript, only Xception[19], MobileNet[20] and ShuffleNet[21] are included in the discussion. This is not enough.

b. I suggest re-organize the paper. In Section 2, it should review papers in the aforementioned research directions. Add a new sub-section such as deep learning method revisit to review three methods: depthwise separable convolution, residual learning and squeeze-and-excitation, since they are the main components in your model.

Response 2: Thanks for the suggestion. We re-write the Related Work part with two subsections Existing Work on Reducing Network Size  and Deep Learning Module Revisit.

We put the details of those three module we used into the Deep Learning Module Revisit  subsection. In subsection Existing Work on Reducing Network Size, we briefly cover model pruning, model compression, weight quantization, and new network structure. It’s hard for us to give a complete literature review in one week, if you have any suggestions, we are happy to add them in the next revised version.

Point 3: The paper title is not appropriate. Since the paper aims to propose a new deep learning structure for real-time applications, the current title does not related to the paper goal at all. In fact, the proposed method is very general and can be applied to any real-time problems, the current title is misleading that the paper is just for blind spot detection problem.

Response 3: Thanks! The network is general, but we still want to emphasize our effort on solving blind spot detection problem. So how do you think this title:

Camera-Based Blind Spot Detection with a General Purpose Lightweight Neural Network

This one emphasize both the solving of real world problem and the general of the model.

Point 4: The experimental results are not convincing enough.

a. In Fig.4, it depicts the blind spot region. However, In Fig.5, sub-figures do not comply with the definition of blind spot in Fig.4. Why blind spot regions can appear in front of cars? In addition, (e) and (f) are examples of CAR class. However, there is no car in the blind spot region. Why they are CAR class?

b. In Table 1 and Table2, we can see the model parameters and the operations of MADD is significantly reduced. However, the inference speed is just 0.001 second. The impact of the proposed method is not significant.

c. It does not compare other model compression methods. Only VGG block is compared in the experiments. This is not convincing enough.

d. In the proposed method, three model settings are used: Sep-Res Block, Sep-SE Block and Sep-Res-SE Block. Which one is better? A discussion section should be provide to compare the results for further conclusion.

Response 4: a. We show the bird view of blind spot region in Fig.4. It’s always on the side of the car. Can you tell me which picture make you think ‘blind spot regions can appear in front of cars’? We will try to eliminate the confusion. (e) and (f) are examples of CAR class in training data without triggering the model. So there is a car in blind spot region, but the bounding box is still green. (g) and (i) are also examples of CAR class in test dataset after triggering the model, so the model can detect the car and change the bounding box to red. We put a short .gif in GitHub, maybe you can check the gif here: https://github.com/placeforyiming/BlindSpotDetection

b.  In the paper of MobileNet, they show the time performance on cpu with milliseconds level. So it’s common to perform in this level. Moreover, the number of parameters is very important not only because inference speed. The GPU on an autonomous car is very limited, since multiple neural networks for different tasks like object detection and road segmentation need to be executed in real time. So less parameters can reduce the scheduled memory cost for this task and save the resource for the other tasks.

c. This paper is aiming at designing network structure to solve real-world problem like blind spot detection. So we didn’t cover the other way to reduce model size. The further research can be done based on our model structure like quantize the weights with the structure in this paper. But we still prefer focusing on one type of method in one paper to avoid confusion.

d. Thanks for the suggestion. Here is the logic of the experiments. Sep is the mainly part we use to reduce the parameters, so all the module contain the Sep part. Res and SE are two modules which may increase the model performance without adding too much cost. So we add those two parts gradually to check if they can provide accuracy increase in our model. 

We expand the Conclusion section to Conclusion and Discussion section. The response here is added into that section to make the result discussion more clear.

Point 5: There are many grammar errors in this manuscript. For example, in the abstract, ‘deep learning method that accurately detect’ should be ‘deep learning method that accurately detects’, ‘a series of experiments is conducted’ should be ‘a series of experiments are conducted’.

Response 5: Sorry for the errors. We correct some errors after checking the grammar again.

Reviewer 3 Report

The paper under review presents a novel solution to the problem of detecting cars in blind spots where drivers cannot see, and therefore give them cues on whether it is safe to change lane. The paper is very well written and self-explanatory, the provided figures are very helpful at that, especially Fig. 3. The results provided seem to be better than current state of the art architectures, such as MobileNet v2.

I am willing to accept this paper for publication, but I have some small concerns and proposals for improvement before that. Most are just typos:

Line: Issue

60: It is said that methods similar to the proposed are difficult to come by and examples shown are too old for computer vision, i.e. [24-26] are older than 10 years or older! A quick Google Scholar search showed some other examples of the same application in the last 4 years, such as "Mono-camera based side vehicle detection for blind spot detection systems" (Baek et al. 2015) among others. I advise the authors to update their previous literature search.

73: becomes > became

88: quick > quickly

89: becomes > became

96: variable "r" is not described anywhere on the text (or did I miss it?)

105: VGG like > VGG-like

Author Response

Point 1: 60: It is said that methods similar to the proposed are difficult to come by and examples shown are too old for computer vision, i.e. [24-26] are older than 10 years or older! A quick Google Scholar search showed some other examples of the same application in the last 4 years, such as "Mono-camera based side vehicle detection for blind spot detection systems" (Baek et al. 2015) among others. I advise the authors to update their previous literature search.

Response 1: Thanks for the reminding! We add new reference paper for bind spot detection in [23-26].

Point 2: 

73: becomes > became

88: quick > quickly

89: becomes > became

105: VGG like > VGG-like

Response 2: Thanks for the correction and sorry for the errors. We correct all those errors in the paper, and correct more errors after checking the whole paper again.

Point 3: 

96: variable "r" is not described anywhere on the text (or did I miss it?)

Response 3: We apologize for the miss of definition. r is the reduction ratio in SE module, this parameter can decide the squeeze level of the module and affect the number of parameters. We add the definition in the paper.

Round 2

Reviewer 1 Report

Can be accepted in its current form.

Reviewer 2 Report

The authors have addressed all the review comments and revised the manuscript accordingly.